# Teacher-Student Semi-supervised Strategy for Abdominal CT Organ Segmentation

Chong Wang, Wen Dong, Daoqiang Zhang, Rongjun Ge

College of Computer Science and Technology, Nanjing University of Aeronautics and Astronautics, Nanjing, China
wangchong9905@163.com

**Abstract.** Semi-supervised abdominal multi-organ segmentation is a challenging topic. In recent years, many methods for automatic segmentation based on fully supervised deep learning have been proposed. However, it is very expensive and time-consuming for experienced medical practitioners to annotate a large number of pixels. Therefore, more researchers focus on semi-supervised learning in abdominal organ and tumor segmentation. In this paper, we adopt a classical Teacher-student semi-supervised strategy to perform the task of abdominal organs and tumor segmentation. Unet is used as the architecture for the segmentation network. Based on the Unet network structure, we add the Inception block and SEBlock to achieve more accurate segmentation. Inception block is its ability to simultaneously capture features at multiple different scales. By introducing SEBlock, the model can better focus on specific information relevant to the task while reducing attention to noise or irrelevant information. Besides, we combine Cross Entropy Loss and Dice Loss as loss functions to improve the performance of our method. We apply a teacher-student model with exponential moving average (EMA) strategy to update the network model parameters. The organs and tumor mean DSC on the public validation set was 85.39%, 18.30% respectively, the organs and tumor mean NSD was 89.36%, 6.44% respectively. And the average running time and the area under GPU memory-time curve 35.54 s, 38175.35.

**Keywords:** Semi-supervised · Abdominal organ segmentation · FLARE2023

## 1 Introduction

The field of abdominal multi-organ segmentation has witnessed significant advancements in recent years, primarily driven by the rise of fully supervised deep learning methods. However, the reliance on fully annotated datasets, which demand considerable time and expertise from medical professionals, has become a bottleneck in further progress[18]. Semi-supervised learning uses existing labeled samples to pseudo-label the remaining unlabeled data, thus mining useful information from unlabeled samples[2,4],which is more practical for the current background.Therefore, more and more researchers begin to pay attention to semi-supervised learning

In this paper, we present a novel approach that leverages the classical Teacher-student semi-supervised strategy to tackle the intricate task of segmenting abdominal organs and tumors. Our methodology is founded on the robust architecture of U-Net[15], a popular choice for image segmentation tasks. Based on the U-Net framework, we introduce Inception block[17] and SEBlock[7] to enable our model to fully understand the belly structure and effectively filter out noise and irrelevant data, thus selectively focusing on relevant information needed for segmentation tasks.

Moreover, we employ a combined loss function approach, merging Cross Entropy Loss and Dice Loss, aimed at augmenting the performance of our method. To facilitate the learning process, we adopt a teacher-student model enriched with an exponential moving average (EMA) strategy for network parameter updates.We evaluated our proposed method on the MICCAI FLARE 2023 challenge dataset, and the experimental results demonstrated the validity of the individual components of our method.

The main contributions of this work are summarized as follows:

- We adopt a two-stage segmentation method, which utilizes a coarse model and a fine model, and adopt a Teacher-Student training strategy of semi-supervised learning to improve the utilization of unlabeled data to achieve robust segmentation results.
- We added Inception blocks and SEBlocks to enhance the ability of the network to capture features at different scales, while enabling the network to learn useful features more efficiently.

## 2    Method

### 2.1    Teacher-Student Model

We propose a method as shown in Fig. 1. We use the coarse model to obtain approximate segmentation results from the input CT scan, and then obtain the region of interest(ROI) coordinates of the abdomen from the coarse segmentation. Then we crop the area, and use the fine model for segmenting, and finally restore the inference results to the original cropped area according to the ROI coordinates. In previous deep learning works, network structure and parameters often need to be adjusted according to practical application. U-Net can achieve good results in most cases. Therefore, we respectively constructed two Unet[15] structural networks with the same architecture and different initial parameters as our coarse model and fine model.

In order to leverage the unlabeled data, we adopt the Mean Teacher model training strategy to achieve semi-supervised learning. Specifically, we first train a teacher model using labeled data and then predict segmentation results for unlabeled data with the trained teacher model as pseudo-labels. Then the student model training in labeled and unlabeled data set with labels and pseudo-labels. The EMA algorithm is used to update the teacher model parameters during the training process.

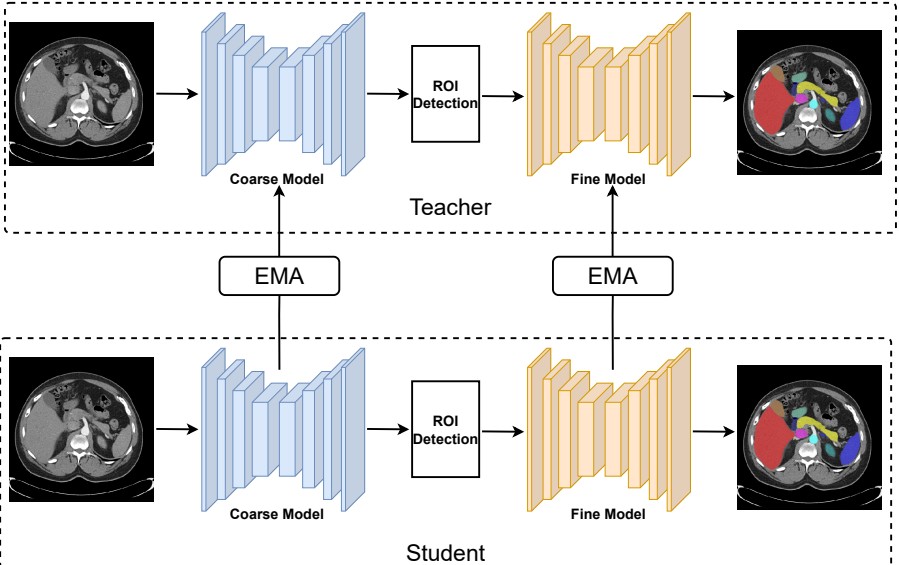

**Fig. 1.** Coarse-to-fine segmentation framework. Coarse and Fine are model inference processes: crop means cutting the approximate position of the organ from the original image according to the result of coarse segmentation, and restore means place the result back to the position before cropping.

## 2.2   Preprocessing

We regroup the 2200 labeled samples and 1800 unlabeled samples to form two datasets. The first dataset containing all 1800 samples is used to train the teacher model, and the second dataset containing only the 4000 labeled and unlabeled samples is used to train the student model for coarse and fine segmentation.

– Reorientation image to target direction.
– For two datasets, we adjust the window width to [-325, 325]. Then the intensities of each CT sample are normalized to have a mean of 0 and a variance of 1 using the individual mean and standard deviation.

## 2.3   Proposed Method

**Network architecture.** We use a UNet structure model as our model as shown in Fig. 2. For the coarse segmentation model, the dimensions are first adjusted to 16 by a 3D convolution. Then, the number of channels after each downsampling is [32, 64, 128, 256], and the input patch size of [160, 160, 160]. For the fine segmentation model, the dimensions are first adjusted to 16 by a 3D convolution. Then, the number of channels after each downsampling is [64, 128, 256, 512], and the input patch size of [192, 192, 192]. The down-sampling is composed of two 3D convolution operations, the first convolution stride is set to 2. The features are

one-half size smaller after the convolution. The up-sampling operation first goes through an interpolation operation, concatenates the features retained during down-sampling through skip connections, and then a 3D convolution operation to adjust the number of channels.

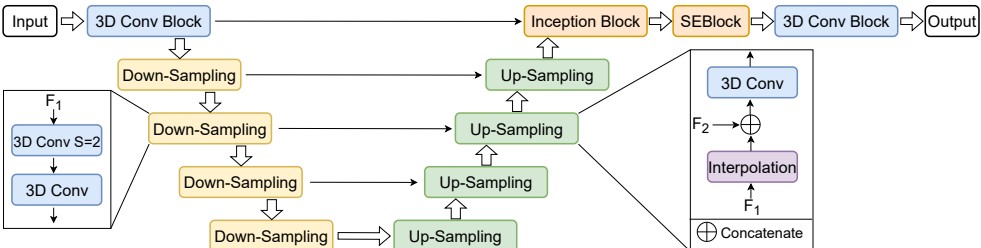

**Fig. 2.** Network architecture. A UNet structure is used and the outputs are used to compute loss. For the coarse segmentation model and the fine segmentation model, we use the same model architecture, but we use different parameters above the model parameters, for the fine segmentation model, we use a deeper model to extract higher dimensional information in the data.

For the segmentation of abdominal organs, which are numerous and vary greatly in size, we propose a method that combines the Inception block and SE-Block with the final segmentation head to enhance the model's feature-capturing ability at different scales. As shown in Fig .3. One of the primary advantages of the Inception block is its ability to simultaneously capture features at multiple different scales. This is highly beneficial for processing images of organs, ranging from small to large, and contributes to enhancing the model's understanding and segmentation performance on abdominal images. The fundamental concept of SEBlock involves adaptively adjusting the weights of each channel to enable the model to more effectively learn useful features. It consists of two main steps: Squeeze (Compression): In this step, SEBlock calculates importance scores for each channel through global pooling operations. This means it considers the average value of each channel across the entire feature map to obtain a weight vector. Excitation: In this step, SEBlock employs a small feedforward neural network (typically a fully connected layer) to learn how to adjust the feature responses of each channel based on their importance scores. This learning process can adaptively increase or decrease the weights of each channel. By introducing SEBlock, the model can better focus on specific information relevant to the task while reducing attention to noise or irrelevant information.

**Loss function.** we use the summation between Dice loss and cross-entropy loss because compound loss functions have been proven to be robust in various medical image segmentation tasks [9].

**Training strategies.** First, train a teacher model on all the labeled data $D_l$. During the training process of the student model, for the labeled data $(x_l, y_l) \in D_l$, input it into the student model obtain $\hat{y}_l$, calculate the loss, and update

the network parameters. For unlabeled data $x_u \in D_u$, input it into the teacher model to obtain pseudo-labels $y_p$, and then input it into the student model to obtain predictions $\hat{y}_u$. Calculate the loss using these predictions and pseudo-labels, and then update the parameters of the student network. The parameters of the teacher model are updated using the Exponential Moving Average (EMA) algorithm. And the student model is used in the inference process.

**Strategies for using partially labeled and unlabeled data.** For the partially labeled data, we don't incorporate pseudo-tags provided by the organizer. For unlabeled data, we input it into the teacher model to obtain pseudo-labels.

**Improve inference speed and reduce resource consumption.** The anisotropic convolution, anisotropic pooling and coarse-to-fine strategy are used to reduce inference time and GPU memory usage.

### 2.4    Post-processing

To avoid the impact of noise, the connected component analysis is used, and we choose the maximum connected component as the final segmentation results.

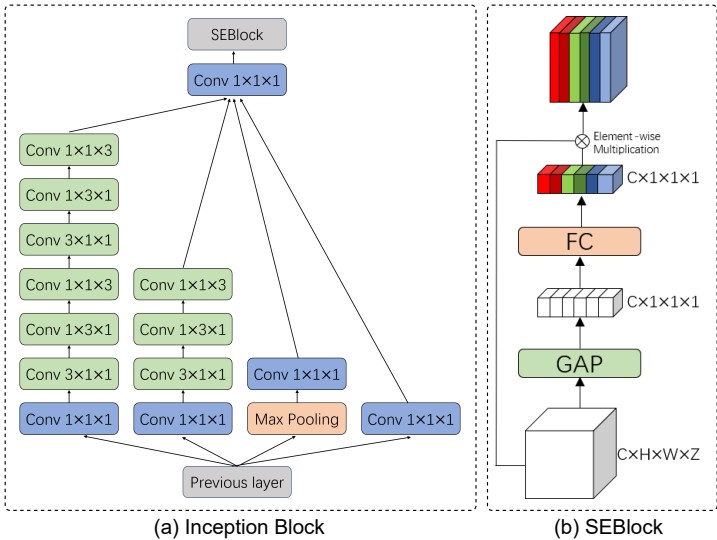

**Fig. 3.** Inception block and SEBlock. Inception block is its ability to simultaneously capture features at multiple different scales. By introducing SEBlock, the model can better focus on specific information relevant to the task while reducing attention to noise or irrelevant information.

## 3   Experiments

### 3.1   Dataset and evaluation measures

The FLARE 2023 challenge is an extension of the FLARE 2021-2022 [11][12], aiming to aim to promote the development of foundation models in abdominal disease analysis. The segmentation targets cover 13 organs and various abdominal lesions. The training dataset is curated from more than 30 medical centers under the license permission, including TCIA [3], LiTS [1], MSD [16], KiTS [5,6], and AbdomenCT-1K [13]. The training set includes 4000 abdomen CT scans where 2200 CT scans with partial labels and 1800 CT scans without labels. The validation and testing sets include 100 and 400 CT scans, respectively, which cover various abdominal cancer types, such as liver cancer, kidney cancer, pancreas cancer, colon cancer, gastric cancer, and so on. The organ annotation process used ITK-SNAP [19], nnU-Net [8], and MedSAM [10].

The evaluation metrics encompass two accuracy measures—Dice Similarity Coefficient (DSC) and Normalized Surface Dice (NSD)—alongside two efficiency measures—running time and area under the GPU memory-time curve. These metrics collectively contribute to the ranking computation. Furthermore, the running time and GPU memory consumption are considered within tolerances of 15 seconds and 4 GB, respectively.

### 3.2   Implementation details

**Environment settings.** The development environments and requirements are presented in Table 1.

**Training protocols.** The Training protocols and details are presented in Table 2 and Table 3

**Data processing.** We regroup the 1800 labeled samples and 2200 unlabeled samples to form two datasets. For two datasets, we adjust the window width to [-325, 325]. Then the intensities of each CT sample are normalized to have a mean of 0 and a variance of 1 using the individual mean and standard deviation.

**Data augmentation.** We adopt the common random enhancement of contrast and random rotation as our data augmentation methods.

**ROI Detection strategy.** We identify organ regions according to the output results of the coarse model, and then the proper RoI can be inferred by calculating the weighted average coordinates and distribution scope of the predicted organ voxels.

**Optimal model selection.** Regarding the selection of the optimal model, we did not set the validation set in the experiment, and we selected the network parameters saved in the last epoch as our optimal model.

## 4   Results and discussion

### 4.1   Quantitative results on validation set

Table 4 shows the results of this work on the validation set. Among the evaluated organs, the liver and spleen demonstrate outstanding segmentation accuracy,

**Table 1.** Development environments and requirements.

| | |
|---|---|
| System | Ubuntu 20.04.1 LTS |
| CPU | Intel(R) Xeon(R) Platinum 8160 CPU @ 2.10GHz |
| RAM | 4×32GB |
| GPU (number and type) | 2*NVIDIA 3090 24G |
| CUDA version | 12.0 |
| Programming language | Python 3.7 |
| Deep learning framework | Pytorch (torch 1.8.2) |
| Specific dependencies | SimpleITK, numpy |
| Code | https://github.com/code-Porunacabeza/flare23 |

**Table 2.** Training protocols for the coarse model.

| | |
|---|---|
| Network initialization | "he" normal initialization |
| Batch size | 2 |
| Patch size | 160×160×160 |
| Total epochs | 150(pretrain on labelled dataset 50 epochs) |
| Optimizer | SGD with nesterov momentum($\mu = 0.99$) |
| Initial learning rate (lr) | 0.01 |
| Lr decay schedule | Halved by 50 epochs |
| Training time | 34 hours |
| Loss function | Cross-entropy loss, Dice loss |

**Table 3.** Training protocols for the fine model.

| | |
|---|---|
| Network initialization | "he" normal initialization |
| Batch size | 2 |
| Patch size | 192×192×192 |
| Total epochs | 200(pretrain on labelled dataset 50 epochs) |
| Optimizer | SGD with nesterov momentum($\mu = 0.99$) |
| Initial learning rate (lr) | 0.001 |
| Lr decay schedule | Halved by 50 epochs |
| Training time | 58 hours |
| Loss function | Cross-entropy loss, Dice loss |

**Table 4.** Quantitative evaluation results.

| Target | Public Validation | | Online Validation | | Testing | |
|---|---|---|---|---|---|---|
| | DSC(%) | NSD(%) | DSC(%) | NSD(%) | DSC(%) | NSD (%) |
| Liver | 96.81 ± 1.84 | 95.28 ± 4.85 | 95.84 | 94.42 | 95.21 | 93.77 |
| Right Kidney | 89.56 ± 20.26 | 87.88 ± 21.03 | 90.02 | 88.27 | 93.42 | 91.56 |
| Spleen | 95.48 ± 7.45 | 95.10 ± 9.78 | 92.66 | 92.23 | 93.74 | 93.09 |
| Pancreas | 78.33± 12.53 | 89.58 ± 12.87 | 76.19 | 87.77 | 77.52 | 87.23 |
| Aorta | 94.95 ± 4.77 | 96.49 ± 5.97 | 95.03 | 96.91 | 95.75 | 97.86 |
| Inferior vena cava | 88.22 ± 14.70 | 88.41 ± 16.29 | 87.38 | 87.45 | 87.46 | 87.89 |
| Right adrenal gland | 80.14 ± 18.61 | 90.19 ± 19.80 | 80.03 | 90.95 | 68.33 | 76.50 |
| Left adrenal gland | 81.95 ± 14.57 | 91.94 ± 14.60 | 79.75 | 89.10 | 75.14 | 82.65 |
| Gallbladder | 76.78 ± 28.12 | 77.66 ± 28.07 | 77.92 | 78.24 | 78.23 | 79.40 |
| Esophagus | 77.10 ± 19.65 | 86.27 ± 20.94 | 77.88 | 87.26 | 82.42 | 91.26 |
| Stomach | 85.70 ± 15.60 | 86.97 ± 15.66 | 86.24 | 87.32 | 87.47 | 88.20 |
| Duodenum | 76.38 ± 12.34 | 89.14 ± 11.52 | 74.74 | 86.84 | 76.11 | 87.85 |
| Left kidney | 89.09 ± 19.72 | 86.78 ± 21.16 | 89.34 | 87.04 | 92.09 | 90.89 |
| Tumor | 18.30 ± 25.14 | 6.44 ± 9.62 | 16.23 | 5.72 | 18.22 | 6.83 |
| Average | 80.59 ± 25.16 | 83.44 ± 27.32 | 79.95 | 82.79 | 79.73 | 82.20 |

with DSC scores exceeding 95%, indicating precise delineation of these critical structures. However, challenges arise in the segmentation of pancreas, gallbladder and tumors, reflected by DSC scores below 80%. This suggests opportunities for further refinement of the segmentation model for improved tumor detection. On average, our method achieves a commendable DSC score of 80.59% ± 25.16% and NSD score of 83.44% ± 27.32% across all organs and tumors, highlighting the effectiveness of our approach.

**Table 5.** Quantitative evaluation of segmentation efficiency in terms of the running them and GPU memory consumption.

| Case ID | Image Size | Running Time (s) | Max GPU (MB) | Total GPU (MB) |
|---|---|---|---|---|
| 0001 | (512, 512, 55) | 29.61 | 3080 | 298264 |
| 0051 | (512, 512, 100) | 16.77 | 3096 | 235758 |
| 0017 | (512, 512, 150) | 36.28 | 3100 | 396605 |
| 0019 | (512, 512, 215) | 30.03 | 3130 | 493118 |
| 0099 | (512, 512, 334) | 32.07 | 3134 | 550114 |
| 0063 | (512, 512, 448) | 43.88 | 3154 | 776612 |
| 0048 | (512, 512, 499) | 49.49 | 3110 | 890207 |
| 0029 | (512, 512, 554) | 58.49 | 3300 | 1056456 |

### 4.2   Qualitative results on validation set

Fig.4 shows some representative good segmentation results. In Case 0029 and Case 0073 examples, our method successfully identified all organs, and the final

predictions are almost the same with the ground truths. The poor segmentation results are shown in Fig.5. In Case 0043 and Case 0059, we can see that there are some under-segmentation and over-segmentation errors in our prediction results. We believe that these poor segmentation results from the ambiguous boundaries of the lesion. Our method performs well in segmenting organs. For healthy organs, we can segment each organ relatively accurately. However, for tumors, our method does not perform well. This may be because tumors are easily confused with organs, leading to the wrong segmentation of organs as tumors.

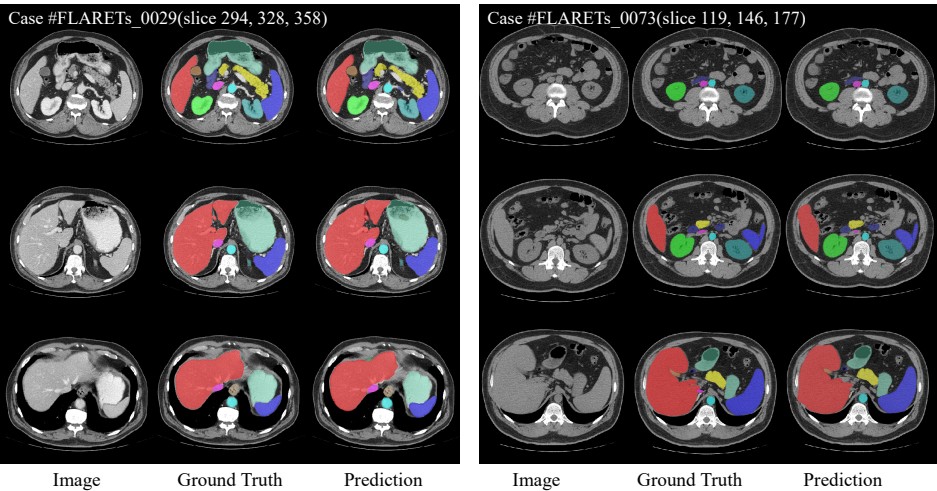

**Fig. 4.** Good segmentation results

### 4.3   Segmentation efficiency results on validation set

In this paper, our segmentation efficiency evaluation is obtained in the development environment shown in the table 1.The segmentation efficiency results are shown in the table 5. The computing resources and time required for samples of different sizes vary. Case 0051 is the fastest at 16.77 s and Case 0029 is the slowest at 58.49 s. This highlights the trade-off between segmentation speed and image complexity, and a similar situation applies in terms of total GPU memory consumption.

### 4.4   Ablation study

Table 6 shows the ablation study results of this work on the validation set. We trained on the first dataset and verified our segmentation results on the validation

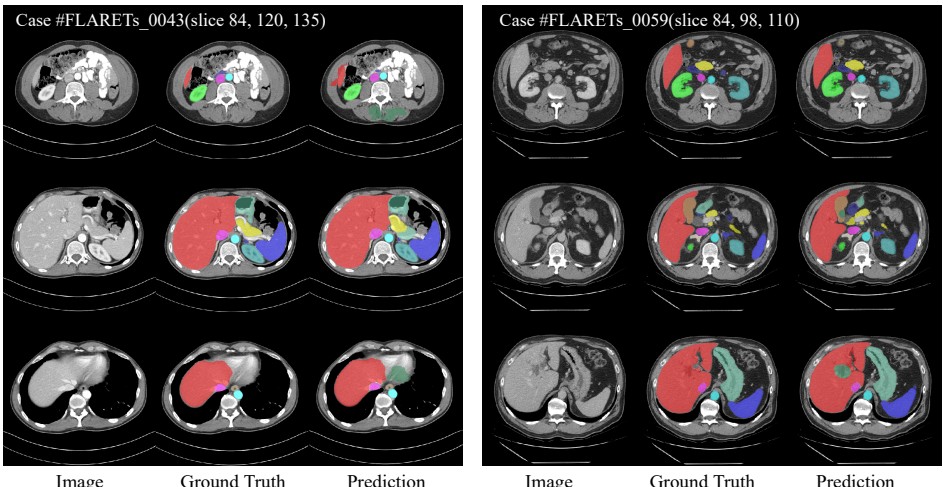

**Fig. 5.** Bad segmentation results

**Table 6.** Ablation study on public validation.

| Target | Without unlabeled | | With unlabeled | |
|---|---|---|---|---|
| | DSC(%) | NSD(%) | DSC(%) | NSD(%) |
| Liver | 96.11 | 92.26 | 96.81 | 95.28 |
| Right Kidney | 88.54 | 86.65 | 89.56 | 87.88 |
| Spleen | 92.46 | 89.67 | 95.48 | 95.10 |
| Pancreas | 79.18 | 89.82 | 78.33 | 89.58 |
| Aorta | 93.81 | 95.20 | 94.95 | 96.49 |
| Inferior vena cava | 88.06 | 88.12 | 88.22 | 88.41 |
| Right adrenal gland | 80.01 | 90.14 | 80.14 | 90.19 |
| Left adrenal gland | 81.82 | 91.92 | 81.95 | 91.94 |
| Gallbladder | 74.58 | 75.97 | 76.78 | 77.66 |
| Esophagus | 76.64 | 85.91 | 77.10 | 86.27 |
| Stomach | 85.35 | 84.90 | 85.70 | 86.97 |
| Duodenum | 74.04 | 89.35 | 76.38 | 89.14 |
| Left kidney | 86.97 | 84.00 | 89.09 | 86.78 |
| Tumor | 11.36 | 2.52 | 18.30 | 6.44 |
| Average | 79.21 | 81.89 | 80.59 | 83.44 |

**Table 7.** Ablation study of SEBlock and Inception Block on public validation training with unlabeled data.

| Variant | Modules | | DSC | |
|---|---|---|---|---|
| | SEBlock | Inception Block | Organs | Tumor |
| Baseline | ✗ | ✗ | 84.63 | 16.77 |
| w/o SEBlock | ✓ | ✗ | 85.07 | 17.62 |
| w/o Inception Block | ✗ | ✓ | 85.12 | 18.39 |
| Full Version | ✓ | ✓ | 85.38 | 18.30 |

set, our method achieves a mean DSC of 79.21% and an NSD of 81.89%. On the second dataset, the semi-supervised training strategy of the teacher-student model is adopted, and both DSC and NSD are improved, achieving mean DSC of 80.59% and NSD of 83.44%. As shown in Table 6, by using unlabeled data, both DSC and NSD are significantly improved and the indicators of all organs improved. It shows that unlabeled data and semi-supervised learning can make the model achieve better performance.

We verified the SEBlock and Inception Block separately using three different network configurations. We summarize the experimental results in Table 7. We used Unet as the Baseline, compared with w/oSEBlock, the DSC of the organs and tumors was improved. This is due to the channel attention of SEBlock, which makes the model focus on the relevant channels that can improve the segmentation performance. The addition of InceptionBolock enables the model to effectively capture image features at different scales and improves the expressive power of the network, making it more adaptive and able to learn complex image features. The DSC of the DSC of the organs and tumors improve to 85.07% and 18.39%, respectively.

### 4.5   Results on final testing set

The test results are shown in Table 4. In the test dataset, we achieved an average DSC of 84.46% and NSD of 88.0% for all organs. At the same time, the average inference time of our method is less than 30s with 44236 GPU memory on average. However, for tumor segmentation, we achieved DSC of 18.22% and NSD of 6.83%. There is still much for improvement.

### 4.6   Limitation and future work

The proposed method works well in most organs. However, the segmentation results of tumors are still unsatisfactory, it has large room to be further improved. Perhaps treating tumor segmentation as a separate task and designing multiple decoders is an effective solution, which is left for future work.

## 5   Conclusion

In this paper, we adopt a Teacher-Student semi-supervised strategy for the abdominal organ segmentation task. We develop and test the whole framework on the FLARE 2023 challenge dataset. The network consists of a coarse segmentation model and a fine segmentation model. We adopt a Teacher-Student semi-supervised learning strategy to leverage a large amount of unlabeled data. We use Unet as the basic network framework and the Inception block[17] and SEBlock[7] combined with the Unet network. The whole framework of our method acquires 79.95% mean DSC and 82.79% mean NSD on the FLARE 2023 challenge validation dataset.

**Acknowledgements** The authors of this paper declare that the segmentation method they implemented for participation in the FLARE 2023 challenge has not used any pre-trained models nor additional datasets other than those provided by the organizers. The proposed solution is fully automatic without any manual intervention. We thank all the data owners for making the CT scans publicly available and CodaLab [14] for hosting the challenge platform.

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
