# OpenReview forum: "Teacher-Student Semi-supervised Strategy for Abdominal CT Organ Segmentation"
_MICCAI.org/2023/FLARE — Submitted to FLARE 2023_

### Official Review · Reviewer_TE2h · 2023-09-19
**Teacher-Student Semi-supervised Strategy for Abdominal CT Organ Segmentation**

**Rating:** 4
**Confidence:** 5

**Review:**

This paper introduces the Inception block and SEBlock to enhance the representative ability of the method. However, it is not clear how the partially labeled data is processed for training. Besides, the teacher-student model distillation and compound loss are common practice in semi-supervised segmentation. The writing also needs to be improved for further consideration.

---

> ### Author Response · Authors · 2023-11-13
>
> Thank you for your valuable review. We have carefully addressed your comments and made the following revisions to enhance the clarity and completeness of our paper:
>
> 1. For the partially labeled data, we don't incorporate pseudo-tags provided by the organizer. For unlabeled data, we input it into the teacher model to obtain pseudo-labels. We have revised the manuscript to provide a more detailed and explicit explanation of how partially labeled data is processed during training.

---

### Official Review · Reviewer_Cg8g · 2023-09-22
**Teacher-Student Semi-supervised Strategy for Abdominal CT Organ Segmentation**

**Rating:** 5
**Confidence:** 5

**Review:**

Pros:

The method proposes a teacher-student model with fine segmentation by extracting ROI regions, introducing Inception block and SEBlock in U-Net to improve the performance, and updating the network parameters through EMA module.

Cons:

2.2 should be 2200 cases of partially labeled data and 1800 cases of unlabeled data.

Lack of clarity in handling partially labeled and unlabeled data

Visualization of qualitative results not displayed as required by the template

It is not clear that Inception block and SEBlock help the model with performance improvement

---

> ### Author Response · Authors · 2023-11-13
>
> Thank you for your detailed review. We have carefully considered your comments and made the following revisions :
> 1. We have updated Section 2.2 to accurately reflect the numbers as 2200 cases of partially labeled data and 1800 cases of unlabeled data.
>
> 2. For the partially labeled data, we don't incorporate pseudo-tags provided by the organizer. For unlabeled data, we input it into the teacher model to obtain pseudo-labels. We have added a relevant content discussion to the revised manuscript.
>
> 3. We re-examined the template and found no requirement for visualization of qualitative results. We have shown two examples with good segmentation results and two examples with bad segmentation results in the validation set.
>
> 4. For the effectiveness of the Inception block and SEBlock, we have conducted additional experiments and included a detailed analysis in the revised manuscript.

---

> > ### Comment · Reviewer_Cg8g · 2023-11-23
> > **Second round of review**
> >
> > The authors have addressed most of the issues and provided detailed explanations to further fulfill publication requirements.
> >
> > However, the authors do not provide details on the strategy for dealing with the use of partially tagged data. Intriguingly, the authors mention that they did not use pseudo-labeling provided by the organizers, so for the 2200 cases of partially labeled data, did the authors use only fully organ-labeled data? And how was the tumor data handled? The authors need to give a more detailed explanation in the implementation details section.

---

### Official Review · Reviewer_Cwv2 · 2023-10-02
**Teacher-Student Semi-supervised Strategy for Abdominal CT Organ Segmentation**

**Rating:** 5
**Confidence:** 4

**Review:**

In this study, a Teacher-Student semi-supervised strategy is employed for the task of abdominal organ segmentation. The network architecture consists of a coarse segmentation model and a fine segmentation model. To leverage a large amount of unlabeled data, a Teacher-Student semi-supervised learning strategy is adopted. The basic network framework utilizes Unet, while incorporating the Inception block and SEBlock with the Unet network.

Cons:
1. It should be "multi-organ" instead of "multiple-organ".
2. The effectiveness of introducing SEBlock and Incepblock have not been explored.
3. The writing need to be improved.

---

> ### Author Response · Authors · 2023-11-13
>
> Thank you for your valuable feedback on our study. We appreciate your insightful comments, and we have addressed the issues you raised in the following manner:
>
> 1. We have revised the term from "multiple-organ" to "multi-organ" throughout the manuscript. Thank you for bringing this to our attention.
>
> 2. To address the concern regarding the effectiveness of introducing SEBlock and Inception Block, we have conducted additional experiments and included a detailed analysis in the revised manuscript.
>
> 3. We have revised the writing to enhance clarity and coherence.

---

> > ### Comment · Reviewer_Cwv2 · 2023-12-01
> >
> > The authors have satisfactorily addressed my concerns.

---

### Official Review · Reviewer_oH6k · 2023-10-03
**Teacher-Student Semi-supervised Strategy for Abdominal CT Organ Segmentation**

**Rating:** 6
**Confidence:** 4

**Review:**

This paper proposes a teacher-student semi-supervised strategy, which uses labeled data to train the teacher model, and uses the pseudo-label data generated by the teacher model and the labeled data to train the student model. The teacher model and the student model update the network parameters through the exponential moving average (EMA) strategy . This method proposes a network framework that introduces SE block and Inception to the traditional Unet, in order to adapt to the identification of changeable abdominal multiple organs. Cons:
The training strategy of partially labeled and unlabeled data are not clearly discussed.
The effectiveness of SE block and Inception module lacks ablation experiments to demonstrate.

---

> ### Author Response · Authors · 2023-11-13
>
> Thank you for your thoughtful review of our paper, and we appreciate your constructive feedback. We have carefully considered your comments and made the following revisions to address your concerns:
> 1. For the partially labeled data, we don't incorporate pseudo-tags provided by the organizer. For unlabeled data, we input it into the teacher model to obtain pseudo-labels.
> 2. In response to your feedback, we have included additional ablation experiments to assess the effectiveness of the SE block and Inception module. The new experimental results and analysis are in the revised manuscript.

---

> ### Comment · Reviewer_oH6k · 2023-11-30
> **Second Round Review**
>
> The ablation study of SEBlock and Inception Block on public validation data is demonstrated in Table 5. Here are some questions: In the training strategy, "Generate tumor pseudo-label on unlabelled data using tumor segmentation model." Does the unlabeled data contain the partially labeled data?

---

### Decision · Program_Chairs · 2023-10-24

Accept